SCIENCE FORUM

# Is preclinical research in cancer biology reproducible enough?

**Abstract** The Reproducibility Project: Cancer Biology (RPCB) was established to provide evidence about reproducibility in basic and preclinical cancer research, and to identify the factors that influence reproducibility more generally. In this commentary we address some of the scientific, ethical and policy implications of the project. We liken the basic and preclinical cancer research enterprise to a vast 'diagnostic machine' that is used to determine which clinical hypotheses should be advanced for further development, including clinical trials. The results of the RPCB suggest that this diagnostic machine currently recommends advancing many findings that are not reproducible. While concerning, we believe that more work needs to be done to evaluate the performance of the diagnostic machine. Specifically, we believe three questions remain unanswered: how often does the diagnostic machine correctly recommend against advancing real effects to clinical testing?; what are the relative costs to society of false positive and false negatives?; and how well do scientists and others interpret the outputs of the machine?

**PATRICK BODILLY KANE AND JONATHAN KIMMELMAN\***

**\*For correspondence:**
jonathan.kimmelman@mcgill.ca

**Competing interest:** The authors declare that no competing interests exist.

In 2012, reports from two major drug companies – Bayer and Amgen – claimed that fewer than a quarter of animal experiments submitted in support of clinical development could be reproduced by in-house researchers at the companies (*Prinz et al., 2011*; *Begley and Ellis, 2012*). These reports seemed to corroborate concerns raised by others around the same time: anti-cancer agents showing promise in animal models had often failed to deliver in clinical trials (*Hay et al., 2014*; *Kola and Landis, 2004*); preclinical cancer studies used methods that did not protect against bias and random variation (*Hirst et al., 2013*; *Henderson et al., 2015*); researchers reported difficulty replicating experiments (*Mobley et al., 2013*); negative preclinical findings often went unpublished (*Henderson et al., 2015*; *Sena et al., 2010*); and studies used too few animals to protect against false positive results (*Button et al., 2013*). Preclinical cancer research seemed to be in the throes of a 'reproducibility crisis'.

Although the Bayer and Amgen articles have garnered over 4,000 citations between them, they did not formalize a definition of reproducibility, or reveal the experiments they tried to repeat, or explain how they tried to repeat them. This raised the question of whether the two studies that questioned the reproducibility of research in cancer were themselves reproducible.

The Reproducibility Project: Cancer Biology (RPCB) set out to address some of the gaps in what we knew about the reproducibility of research in cancer biology. Building on a previous effort that attempted to replicate 100 studies in experimental psychology (*Open Science Collaboration, 2015*), a team of researchers at the Center for Open Science used an explicit method to sample 50 impactful publications in cancer biology (*Errington et al., 2014*). Then, working with Science Exchange (an organization that helps organizations to outsource R&D), they formalized their definitions of reproducibility, selected individual experiments within each of the 50 publications, and pre-specified protocols for repeating these experiments. These protocols were then peer reviewed by *eLife* and published as Registered Reports (https://www.cos.io/initiatives/registered-reports). An important part of the project was that data collection could not start before the Registered Report was accepted for publication. In the end, due to a combination of technical and budgetary problems, Registered Reports were published for 29 of the 50 publications.

The experiments were then performed, usually by contract research laboratories or core facilities at universities, as specified by the protocols in the relevant Registered Report. The RPCB team then wrote a Replication Study that contained: (a) the results of these experiments; (b) a discussion of how the results/effects compared with the results/effects reported in the original research article; and (c) a meta-analysis that combined the data from the original experiment and the replication. This Replication Study was peer-reviewed (usually undergoing revision) and then published. By the end of the project 17 Replication Studies had been published; Replication Studies were not published for 12 of Registered Reports due to technical and/or budgetary problems, but the results from eight partially completed replications have been published (*Errington et al., 2021a*; *Pelech et al., 2021*). The RPCB project has also published a paper containing a meta-analysis of all the replications (*Errington et al., 2021b*) and, separately, a paper that describes how the project was carried out and some of the challenges encountered during it (*Errington et al., 2021c*). The meta-analysis covers a total of 158 different effects from 50 experiments in 23 papers.

For each Replication Study the *eLife* editors handling the study added an assessment of the success or otherwise of the replication in the form of an Editors' Summary. According to these summaries: five of the replications reproduced important parts of the original research articles; six reproduced parts of the original research articles but also contained results that could not be interpreted or were not consistent with some parts of the original research article; two could not be interpreted; and four did not reproduce the parts of the original research articles that they attempted to reproduce.

In what follows, we consider some of the scientific, ethical and policy implications of the project, and discuss three key questions about reproducibility that are left unresolved.

## Preclinical studies as diagnostic machines

To understand the implications of the RPCB it is important to appreciate two ways that the meaning of reproducibility in cancer biology differs from that in other areas in which large-scale reproducibility projects have been carried out (that is, psychology [*Open Science Collaboration, 2015*], experimental economics [*Camerer et al., 2016*], and the social sciences [*Camerer et al., 2018*]). First, research in psychology, economics and the social sciences is usually conducted on humans and the goal is to identify causal relationships that generalize to other humans. In contrast, research in cancer biology is conducted in tissue cultures and non-human animals, and the goal is to identify causal relationships that generalize to living humans. This means that the claims about causal relationships made by cancer biologists are always dependent on a more extensive set of assumptions – which are fallible – about the relationship between experimental systems and real-world settings.

Second, research in psychology and economics is generally geared toward generating, validating and comparing theories about causal relationships (for example, testing whether the violation of social norms leads to more norm violation). The purpose of replication in these fields, therefore, is to determine how much of what we think we know about these causal relationships is true. Pre-clinical cancer research, by contrast, is part of a broader enterprise tasked with finding treatments for diseases. The primary point of this research is to help decide which claims should be advanced into treatments, and which novel treatments should be advanced to further evaluation and rigorous testing in clinical trials. Thus, the replication question is a practical one: do contemporary research practices efficiently prioritize potential strategies for development and clinical testing, given the limits of existing model systems? (We appreciate that some biomedical research is pursued more in the spirit of fundamental inquiry rather than application but, nevertheless, we maintain that medical objectives dominate the motivations of private and public research sponsors).

In clinical testing, as many as 19 of 20 cancer drugs put into clinical development never demonstrate enough safety, efficacy or commercial promise to achieve licensure (*Hay et al., 2014*). Even those few that run the gauntlet from trialing to license often show underwhelming efficacy in trials (*Davis et al., 2017*). These failures exact enormous burdens on both the non-human animals that are sacrificed for this effort and the many patients who volunteer time, welfare and good will in offering their bodies and tissue samples for clinical trials. Futile research efforts also divert talent, patients and funding from more promising ones, often at the expense of the taxpayer in the form of drug reimbursement. The initial response of many researchers to the COVID-19 pandemic clearly showed how poorly planned, executed and reported research efforts

can divert attention and resources from more productive endeavors (*London and Kimmelman, 2020*).

As one of us (JK) and Alex London have argued, many of these failures reflect the intrinsic challenges of conducting research at the cutting edge of what we understand about disease (*London and Kimmelman, 2015*). The failure of numerous drugs inspired by the amyloid cascade hypothesis, for example, in part reflects uncertainties about the pathogenesis of Alzheimer's disease. When translation failures are due to theoretical uncertainties like this, properly designed and analyzed trials provide important feedback on experimental models and pathophysiological theories (*Kimmelman, 2010*). These types of 'failures' – which are natural and informative – are not the type of errors that the RPCB has (or should) set out to address.

The RPCB is instead concerned with the extent to which clinically interesting causal relationships discovered in laboratories are reproducible using the same experimental systems. Concluding that causal relationships are real when they are not represents a potentially 'unforced error' in drug development. If a treatment is advanced to clinical research based on such an error, medicine is deprived of an opportunity to develop an effective treatment and, furthermore, the research enterprise is deprived of the opportunity to learn something about the generalizability of our models and theories. These unforced errors are a sort of tax or friction that makes the research process significantly less efficient than it could be. Many of the issues that lead to such errors – haste, poor study design, biased reporting – are relatively easy and cheap to fix. And given the prevalence of preclinical studies that do not reproduce according to Amgen and Bayer, the gains of reducing such unforced errors could be substantial.

These unforced errors can come in two varieties. False positives occur when research communities mistakenly conclude that a clinically promising causal relationship exists when it does not. Projects looking into the reproducibility of research tend to focus on false positives because they are mostly interested in how valid our knowledge in a given field is. False negatives occur when a real and potentially impactful causal relationship is assumed to be too weak to be clinically promising, or the relationship is not observed in the first place. These tend to be ignored in reproducibility projects. However, in the context of preclinical cancer research, false negatives require serious consideration because the cost to society of missing out on impactful cancer treatments can be considerable, especially given that clinical development exists to cull false positives but can do little about false negatives.

One can think of the basic and preclinical research enterprise as a sprawling, multiplex diagnostic machine that helps researchers to decide if a clinical hypothesis should proceed to more rigorous development, such as a clinical trial. Clinical hypotheses are fed into the machine, which runs a series of experiments, and the results of these experiments are assembled into research articles, which are then submitted to journals. If the individual experiments within a research article converge on the same translational claim (which may be a claim about a particular molecule, or a claim about strategy), and if the article reports a large effect for a phenomenon judged by experts to be relevant to translation, the article is accepted for publication in a high-impact journal. However, if the individual experiments point in inconsistent directions, or if the effects are small, or if quality control flags a study, the article will not be published in a high-impact journal. The decision to proceed to a clinical trial (or to intensify research) will typically be made based on a small number of research articles (that is, on a few outputs of the diagnostic machine), other forms of evidence (such as safety information and trials involving related treatment strategies), and other extra-scientific considerations (such as commercial potential).

Some clarifications and provisos are needed before we proceed with our analysis. First, we acknowledge that analogizing cancer biology research as a diagnostic machine does not accommodate the spirit and goals of all the publications included in the RPCB. Like all models, ours strips away complexity to bring essential elements of a system to the fore (*Borges, 1946*). Nevertheless, as noted previously, every sponsor funding these efforts was probably motivated by the prospects of treating or preventing cancer. Even the most basic findings of studies in the RPCB have some prospect of being advanced in some fashion to clinical applications. Second, we recognize that positive outputs of the diagnostic machine will be used differently. In some cases they might be grounds to launch a clinical trial. In other cases, they will promote a clinical hypothesis to further work on a critical path towards translation (*Emmerich et al., 2021*) – perhaps by supporting the development of more pharmacologically attractive compounds that target the causal process described in a research article.

## Box 1. A primer for assessing the performance of the diagnostic machine

At a high level we can think of diagnostic testing in terms of six factors.

1. Base rate: this is the prevalence or 'prior' of true clinical hypotheses among those fed into the diagnostic machine. This will vary by field: the base rate is very low in areas like Alzheimer's disease, where we have few well-developed clinical hypotheses. However, the base rate is likely higher in, for example, hemophilia, where causal processes of disease are well understood.

2. Sensitivity: how frequently the diagnostic machine produces a positive output when it is testing a real causal relationship.

3. Specificity: how frequently the diagnostic machine produces a negative output when it is testing a null effect.

4. Positive Predictive Value (PPV): the frequency of real causal effects amongst positive outputs of the diagnostic machine.

5. Negative Predictive Value (NPV): the frequency of null effects amongst negative outputs of the diagnostic machine.

6. Likelihood ratio: this is the ratio of the probability the diagnostic machine produces a particular result when a real causal effect is being tested to the probability the diagnostic machine produces that same result when a null effect is being tested (*Goodman, 1999*). When we restrict the diagnostic machine to producing either a 'positive' or 'negative' result, there is a likelihood ratio corresponding to each. The positive likelihood ratio corresponds to the sensitivity divided by one minus the specificity, while the negative likelihood ratio is one minus the sensitivity divided by the specificity. A positive likelihood ratio of 1.0 indicates that a positive result is just as likely to occur when testing a null effect as a real causal effect. The higher the positive likelihood ratio, the more likely a positive output from the diagnostic machine indicates a true clinical hypothesis.

There are two points worth making about these numbers. First, the base rate, sensitivity and specificity determine the PPV and NPV (see *Box 2*): one can think of the PPV and NPV as properties that emerge when a test with a given sensitivity and specificity is applied to a population of hypotheses with a particular base rate of real causal effects. Second, although there is no mathematical requirement that sensitivity and specificity be linked, in practice there is often an inverse relationship between the two. For example, most scientists require a p value of less than 0.05 in order to reject a null hypothesis. Using a p value of 0.10 instead would make it easier to get a significant finding, thus raising the sensitivity. However, using a p value of 0.10 would also increase the number of false positives and lower the specificity.

Furthermore, in our analogy, the outputs of the diagnostic machine are to be understood not as epistemic outputs (that is, positive outputs truly signal a hypothesis is promising; negative outputs signal the reverse). Instead, they are to be understood as sociological outputs, with a positive output signaling that various expert communities regard a hypothesis as worthy of further development. In this way, our analogy regards the RPCB as determining whether repetition of key experiments in original research articles would have produced results consistent with previous ones that generated a 'buzz' in the cancer biology community. In a well-functioning research system, sociological truths will track epistemic truths. However, the former will also be informed by non-epistemic variables, including views about clinical need. In the next sections, we consider what the RPCB results tell us about the performance of this diagnostic machine; *Box 1* provides a primer on the terminology that will be used in our analysis (such as Positive Predictive Value and Negative Predictive Value).

### The Positive Predictive Value is low for preclinical cancer studies

As specified by their sampling approach (and like other large-scale replication studies), the RPCB only selected research articles that the diagnostic machine had labeled as 'very positive'. All the papers selected for the project had received high numbers of citations, meaning that they had made an impact among cancer researchers: moreover, many of them had been published in high-profile journals such as *Cell*, *Nature* and *Science*, although this was not one of the selection criteria. Therefore, the principal diagnostic property directly assessed by the project was the Positive Predictive Value, which is defined as the number of outputs that replicated (true positives) divided by the total number of outputs tested.

As mentioned previously, only 17 out of the 50 originally planned Replication Studies were completed, with varying degrees of success: five reproduced important parts of the original research articles; six produced equivocal results; two could not be interpreted; and four did not reproduce the experiments they attempted to reproduce (an additional six replication reports were only partially completed). For simplicity we assume that 50% of the equivocal results would regress on repetition, leading the diagnostic machine to recommend advancing with three of these studies. Adding these three studies to the five studies that reproduced important parts of the original research articles, and dividing by 17 replication attempts, gives us an initial estimate of 47% for the Positive Predictive Value of the diagnostic machine. To put this number in context, replication rates for articles in three leading journals in psychology were (depending on how you count) 40% (*Open Science Collaboration, 2015*); the figure for articles in two leading journals in economics was 66% (*Camerer et al., 2016*); and for articles in the social sciences published in *Nature* and *Science* the figure was 67% (*Camerer et al., 2018*).

It is more likely than not, however, that an estimate of 47% for the Positive Predictive Value is charitable. As mentioned previously, according to the Editors' summaries, 2 of the 17 published Replication Studies reported findings that could not be interpreted (due to unexpected challenges in repeating the original experiments), and 33 studies were abandoned due to cost overruns, difficulties in securing research materials, or a lack of cooperation from the original authors. The RPCB team found that the original authors were bimodal in their helpfulness when it came to

providing feedback and sharing data and materials: 26% were extremely helpful, but 32% were not at all helpful or did not respond (*Errington et al., 2021c*).

It might be tempting to view the 33 abandoned efforts as uninformative, but we may be able to actually learn from them because it is likely that the 17 published studies are biased in favor of work that is reproducible. Laboratories that were more confident about the reproducibility of their original publications, or more fastidious with their record keeping, might be more likely to cooperate with the RPCB than laboratories that were more doubtful or less fastidious (although the RPCB team did not observe a relationship between material sharing and replication rates; *Errington et al., 2021b*).

The RPCB also selected experiments that did not rely on unusual samples or techniques, which probably increased the chances of successful replication: experiments that require unusual samples or techniques are, it seems to us, more likely to require more attempts to get them 'to work', and therefore more likely to be prone to the problem of 'researcher degrees of freedom' (*Simmons et al., 2011*).

The problems the RPCB team experienced in terms of important information not being included in the original research articles, or the original authors not sharing data and/or reagents (*Errington et al., 2021c*), also points to a diagnostic machine whose workings are often opaque and that leaves a very patchy audit trail for its outputs.

In medicine, the number of people dropping out of a clinical trial (a process called attrition) is often regarded as a useful piece of information and is used when evaluating the results of the trial in 'intention to treat' analyses. Based on what has been reported thus far by the RPCB, eight replication studies produced results consistent with original research articles. The remaining 42 replication studies did not, thus providing a lower bound on the Positive Predictive Value of 16% – an estimate that is not far off from the proportion of studies Amgen reported reproducing years ago (*Begley and Ellis, 2012*).

As mentioned previously, the meta-analysis covered a total of 158 different effects. Most of the original effects were positive effects (136), and for these the RPCB team found that the median effect size in the replications was 85% smaller than the median effect size in the original experiments; moreover, in 92% of cases the effect size in the replication was smaller than in the original. If the original publications represented unbiased

estimates of real effects, one would expect the replications to regress towards smaller effect sizes as often as they drifted towards larger effect sizes. Similar regression has been seen in other large-scale replication projects and noted by other commentators (*Colquhoun, 2014*).

The poor reproducibility of the individual experiments should influence our interpretation of the overall study results. According to our estimates, the abstracts of 13 of the 17 Replication Studies (that is, 76% of the studies) noted at least one experimental claim that did not reproduce the original effect (meaning that, had replication studies been submitted as original publications, their 'stories' would have been less coherent, putting them at higher risk of rejection by a journal).

## Against despair, part I: Negative Predictive Value

So far, things are not looking good for our cancer biology diagnostic machine, but there is more to diagnosis than the Positive Predictive Value. The Negative Predictive Value is also important, given that any positive output will be further tested in clinical trials, but negative outputs will be turfed. Unfortunately the RPCB (and similar projects) only tried to reproduce papers asserting positive causal relationships, thus providing no information about how often the diagnostic machine falsely labels submissions as 'negative'. (Although the 158 effects analyzed included 22 null effects, these were embedded in papers that contained mostly positive results, so they are unlikely to be representative of the broader population of negative results.) Given this limited information, it may be that the low PPV is balanced out by a high NPV, suggesting that at the very least we are not missing out on many potential valuable clinical hypotheses. On the other hand, the NPV may be as bad or worse than the PPV. Without further study of negative outputs from the diagnostic machine, we simply cannot know how reproducible they are.

Our diagnostic machine might return a falsely negative output for a number of reasons: an individual experiment might not be implemented using proper technique; a key individual experiment might be underpowered; a closely related rival clinical hypothesis might be accepted instead because of a biased individual experiment; or a journal might reject an article reporting a valid clinical hypothesis. To our knowledge there have been no attempts to assess the NPV of a field of research, possibly because many negative outputs

are never written up, never mind submitted to a journal; indeed, according to one estimate, only 58% of animal studies are eventually published (*Wieschowski et al., 2019*). It might be possible to avoid this problem by asking a number of laboratories to blindly repeat a series of original research articles that support clinical hypotheses that have since been confirmed: the proportion of replication attempts that 'fail' would provide some insight into the NPV. However, this approach would require significant resources, and perhaps there are better ways to probe the NPV of preclinical research in cancer biology and give us a more complete understanding of the systems we currently rely on to prioritize clinical hypotheses for further development.

## Against despair, part II: Decision rules

A second argument against despair concerns what are sometimes called 'decision rules'. In diagnosis, not all errors have the same practical or moral significance. The statistician Jerzy Neyman, writing in the 1950 s, offers the example of X-ray screening of healthy persons for tuberculosis. A false positive might cause anxiety while that individual waits for the results of further tests. However, a false negative means the person is denied an opportunity to undergo treatment early in a disease course (where management is more effective) and is likely to unwittingly spread the disease (*Neyman, 1950*). Decision rules illustrate one of the important ways that moral and social propositions are embedded within concepts and interpretations of reproducibility. Indeed, a richer criterion for 'reproducibility' would be whether a given experiment (or set of experiments), when repeated, provides similar levels of support for decisions that an original experiment (or set of experiments) aimed at informing.

What sort of decision rule is appropriate for weighting up false positives and false negatives from the diagnostic machine? As discussed above, preclinical research forms an intermediate step between theory and clinical trials. So long as we have mechanisms for intercepting false positives elsewhere in the research enterprise (ideally, before clinical trials begin), damage will be limited. Moreover, some of the costs associated with redesigning the diagnostic machine so that it produces fewer false positives are likely to be significant: for example, replacing our religious devotion to a *p*-value of 0.05 with an even more demanding value, such as 0.005 (*Benjamin et al., 2018*), would require researchers to use much

larger sample sizes in their experiments. This might be fine for experiments in psychology and other areas that can use something like Amazon's Mechanical Turk to recruit participants. In medicine, however, the total volume of human talent, tissue samples and non-human animals available to researchers is limited, so larger sample sizes would mean testing fewer clinical hypotheses, thus limiting our ability to scan the vast landscape of plausible hypotheses.

On the other hand, false negatives mean that populations are deprived of access to a potentially curative therapy, at least until errors are corrected. False negatives might take longer to correct because scientists have less motivation to repeat negative experiments. False negatives will be especially costly in underfunded research areas like neglected diseases and pediatric cancers, since errors will not be quickly corrected by competing labs conducting research in parallel. In well-funded areas where there are many research teams (such as, say, lung cancer), these costs may be small, as competing laboratories pick up on hypotheses rejected by other laboratories. So our decision rule – and the level of reproducibility we should demand for cancer preclinical research – depends on a set of moral and sociological conditions that are outside the scope of the RPCB.

## Against despair, part III: Beliefs

Finally, diagnostic machines are simply tools to inform expert beliefs. Ultimately their utility can only be judged on how experts use them. For all 50 research articles identified by the RPCB, the diagnostic machine had originally recommended further development. Many outputs from the diagnostic machine appear, in retrospect, to have been biased in favor of accepting novel clinical hypotheses. This bias is undesirable from an ethical standpoint because non-human animals and scientific effort have been wasted on spurious hypotheses, but what matters most from the standpoint of science is that experts interpreted positive outputs from the diagnostic machine – that is, articles in high-impact journals – correctly and applied these outputs appropriately.

There are many reasons why experts contemplating studies like those in the RPCB might have interpreted and acted on the original research articles differently. The prior probability of a clinical hypothesis in one of the original research articles being true is likely to have depended on previous knowledge in that area of cancer biology. In cases where prior probabilities were

higher (in diagnostics, the equivalent of having characteristic symptoms or a higher disease prevalence), a positive output may be sufficient to launch a clinical trial, whereas in cases where prior probabilities were lower, further evidence would be required. Experts might also interpret and act on positive outputs differently because of the methods used. Just as a positive PCR test on a patient might be viewed more skeptically if samples have not been fastidiously protected from contamination, an expert might ask for more evidence if the positive results in a research article were obtained with a method that is known to be temperamental or fallible. On this view, the decision of Amgen and Bayer to replicate preclinical work in house suggests that companies are well aware of the unreliability of many preclinical reports in the published literature.

The RPCB, like most other empirical studies of reproducibility, focused on the 'material' dimensions of reproducibility: the adequacy of the reagents and methods used and the completeness of the reporting in the original research articles, and the ability of independent scientists to implement the protocols described in these articles and obtain statistically similar results. Yet reproducibility has a cognitive dimension as well, namely the ability of experts to use evidence from various sources to predict if, how and to what extent experimental results will generalize. Little is understood about this aspect of reproducibility – and it may matter as much or more than the material dimensions. How well are competent experts able to read the methodology section of an article, and correctly infer that they can implement the exact same protocol in their own lab? How well can experts assess whether, if they implement the exact same protocol, they can obtain results that are statistically consistent with the original research article? How well can experts anticipate the extent to which variations in experimental conditions will extinguish previously detected cause and effect relationships? And how well can experts judge the extent to which phenomena detected under closely-controlled laboratory conditions will be recapitulated in the wilds of a clinical trial?

The little we know about this aspect of reproducibility provides mixed signals. On the one hand, forecast studies carried out by one of us (JK) and colleagues have found that: (i) preclinical researchers are unable to predict whether replication studies will reproduce experiments (*Benjamin et al., 2017*); (ii) cancer experts perform worse than chance in predicting which treatments tested in randomized trials will show

## Box 2. Calculating sensitivity and specificity

We can write the Positive Predictive Value (PPV) and the Negative Predictive Value (NPV) in terms of the sensitivity, specificity and base rate (BR) as follows:

$$PPV = \frac{BR x Sensitivity}{BR x Sensitivity + (1-BR) x (1-Specificity)}$$

and

$$NPV = \frac{(1-BR) x Specificity}{(1-BR) x Specificity + BR x (1-Sensitivity)}.$$

We can re-arrange these two equations to obtain the following expressions for the sensitivity and specificity:

$$Sensitivity = \frac{PPV x (1-BR) x (1-Specificity)}{(1-PPV) x BR}.$$

and

$$Specificity = \frac{NPV x BR x (1-Sensitivity)}{(1-NPV) x (1-BR)}.$$

We can then plug the expression for the specificity into the expression for sensitivity to obtain the following expression:

$$Sensitivity = \frac{PPV x (1-NPV) x (1-BR) - PPV x NPV x BR}{BR x (1-PPV) x (1-NPV) - PPV x NPV x BR}.$$

It is also possible obtain a similar expression for the specificity. The sensitivity and specificity can then be converted to likelihood ratios using the definition in **Box 1**.

efficacy (**Benjamin et al., 2021**); (iii) the predictions of cancer experts about trial outcomes are also heavily influenced by the format in which evidence is presented to them (**Yu et al., 2021**). However, these studies all have limitations and findings from other teams are more sanguine. For example, some studies suggest that expert communities are – in the aggregate – able to pick out which studies replicate in psychology (**Dreber et al., 2015**) and the social sciences (**Camerer et al., 2018**), and experts appropriately update their beliefs about an experimental phenomenon on seeing new and more powerful evidence (**Ernst et al., 2018**).

Future studies of reproducibility in the clinical translation process will need to use qualitative research and methods from decision science to characterize how decisions are made about starting clinical development, what experts believe about the prospects of successful translation, what sorts of evidence they rely on to form these beliefs, and how these beliefs are aggregated to arrive at the institutional decision of the sponsor.

## Conclusions and next steps

For the sake of simplicity, let us assume that, across the field of cancer research, the base rate is 10%: that is, the prior for credible cancer research hypotheses being sufficiently true is 10%. Here the phrase 'sufficiently true' means that a hypothesis is close enough to being true that any adjustments that need to be made to the hypothesis can be made during clinical development. Let us also assume that 99% of negatives are true negatives, so the NPV is 99%. What then are the sensitivity and specificity of the diagnostic machine?

If we assume that the PPV is the higher of the two estimates we derived earlier, 47%, then the sensitivity of the diagnostic machine is 92% and the specificity is 88% (see **Box 2** for the relevant formulae). This translates to a likelihood ratio for all positive preclinical studies of 8: that is, the diagnostic machine is eight times more likely to recommend clinical testing for real causal effects than for null effects. However, if we use the 'intention to treat' principle and adopt the lower

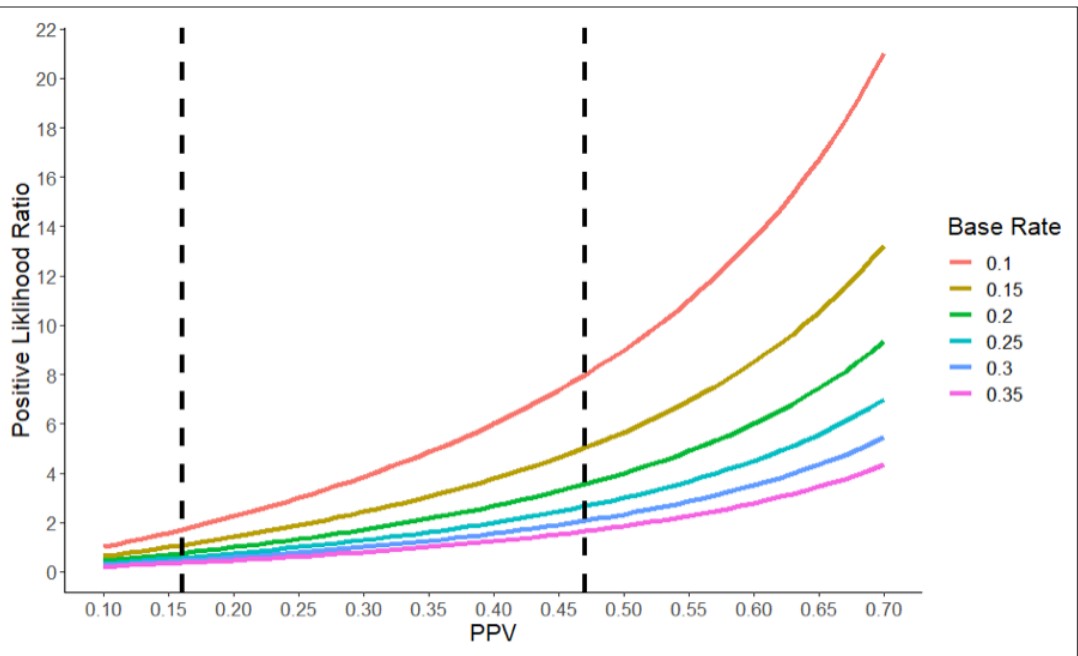

**Figure 1.** The positive likelihood ratio as a function of the Positive Predictive Value (PPV) and the base rate. The positive likelihood ratio (y-axis) increases with the PPV (x-axis) for a given value of the base rate (see color code). However, for a given value of the PPV, the positive likelihood ratio decreases as the base rate increases. The vertical dashed lines represent the two estimates of the PPV (16% and 47%) we derived for the RPCB. Typically, a positive likelihood ratio between 1 and 2 is considered weak evidence, while ratios between 2 and 10 constitute moderate evidence, and ratios higher than 10 constitute strong evidence. A ratio of less than one indicates that a test is actively uninformative.

estimate of the two estimates for PPV, 16%, the positive likelihood ratio drops to 1.7, driven by a sharp fall in the specificity.

Likewise, the likelihood ratios would be very different if we were to make different assumptions about the base rate and the NPV (see *Figure 1*). For example, if we were to assume the base rate of true hypotheses is 30% and the PPV is 47%, then the positive likelihood ratio drops to 2, which constitutes extremely weak evidence. Even with the most generous set of numbers, preclinical studies of the type sampled in the RPCB (that is, studies that receive high numbers of citations) provide only moderate evidence that a clinical hypothesis is true.

Several practical points about clinical translation follow from this analysis. First and most obviously, preclinical studies should never be interpreted in isolation from other theory and evidence. Instead, decision-makers should actively seek corroboratory and disconfirmatory evidence and interpret any claim in a preclinical study against what was known previously.

Second, as a general rule, positive preclinical publications should be understood in light of their role as providing exploratory evidence

for future clinical work. This means we should understand the importance of both winnowing the field of potential hypotheses while also not removing any plausible hypotheses from consideration. Societies committed to improving patient outcomes and using healthcare resources wisely need to maintain strong mechanisms for subjecting clinical hypotheses to a re-test before they are advanced into clinical practice. Emmerich and co-authors offer a pathway for assessing new strategies for clinical development that is informed by 'validity threats' in preclinical research (*Emmerich et al., 2021*). There have also been call for researchers to label preclinical studies as exploratory, and for findings and hypotheses to be subject to confirmatory testing using principles of pre-specification (along the lines of registered reports) before they are advanced into clinical development (particularly if a hypothesis is likely to entail risk and burden [*Mogil and Macleod, 2017*; *Dirnagl, 2019*; *Drude et al., 2021*]).

Regulators, sponsors and ethics committees should expect that key preclinical studies be replicated in confirmatory studies where hypotheses and protocols are pre-specified and

pre-registered (*Kimmelman and Anderson, 2012*) and proper statistical and experimental methods (such as randomization) are used. Governments should maintain strong pre-license drug regulatory standards, which provide powerful incentives for companies to confirm clinical hypotheses. Many recommendations in clinical practice guidelines are based on mechanistic evidence and expert judgment. Moreover, the journey from bench to bedside is becoming shorter with the emergence of precision medicine, techniques such as patient-derived xenografting (*Kimmelman and Tannock, 2018*; *Yu et al., 2021*; *Byrne et al., 2017*), and a weakening of drug regulation. RPCB findings, and our own studies of medical scientist expert prediction (*Benjamin et al., 2017*), provide grounds for believing that many such medical practices may sometimes result in costly and needlessly burdensome clinical practices.

Third, a complete understanding of the previous point requires accessing more information on the NPV and base rate of true hypotheses under consideration. Without this information we can only guess at whether the diagnostic machine is optimally tuned to maximize societal gains against the non-human animals, patients, and human capital we invest in operating it. Creativity and a great deal of cooperation from labs will likely be necessary for estimating these numbers.

Finally, we need to expand our inquiry into reproducibility by studying its non-material dimensions in greater detail. As we have tried to emphasize, defining and understanding reproducibility requires grappling not merely with what is done with pipettes and Eppendorf tubes, but also with the psychology of expert inference and decision-making, and how these judgments coalesce within research communities. It will require more thinking about ethical and pragmatic judgments embedded within decision rules. It will also require evidence and analysis concerning the efficiency of existing research systems, and the moral trade-offs between false positivity and false negativity. Though the RPCB has brought us much closer to knowing the extent to which we can trust conclusions in cancer preclinical studies, many evaluative judgments regarding the cancer biology research enterprise will have to await further scientific, sociological and moral inquiry.

### Acknowledgements
We thank Selin Bicer for research assistance. This work was supported by funding from a large-scale applied research project grant from Genome Quebec, Genome Canada, the Government of Canada.

**Patrick Bodilly Kane** is based in Studies in Translation, Ethics and Medicine, Biomedical Ethics Unit, McGill University, Montreal, Canada
🔗 http://orcid.org/0000-0003-1050-570X
**Jonathan Kimmelman** is based in Studies in Translation, Ethics and Medicine, Biomedical Ethics Unit, McGill University, Montreal, Canada
jonathan.kimmelman@mcgill.ca
🔗 http://orcid.org/0000-0003-1614-6779

*Author contributions:* Patrick Bodilly Kane, Conceptualization, Data curation, Formal analysis, Funding acquisition, Project administration, Visualization, Writing – original draft, Writing – review and editing; Jonathan Kimmelman, Conceptualization, Data curation, Formal analysis, Funding acquisition, Project administration, Visualization, Writing – original draft, Writing – review and editing

*Competing interests:* The authors declare that no competing interests exist.

### Funding

| Funder | Grant reference number | Author |
|--------|------------------------|--------|
| Genome Quebec, Genome Canada | | Jonathan Kimmelman |

The funders had no role in study design, data collection and interpretation, or the decision to submit the work for publication.

### Decision letter and Author response
Decision letter https://doi.org/10.7554/eLife.67527.sa1
Author response https://doi.org/10.7554/eLife.67527.sa2

### Data availability
No data was generated for this work.

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
