## [Decision Letter]

Note: This article underwent further editing and revision after it had been revised in response to feedback from the peer reviewers.

Thank you for sending your Feature Article "Is cancer biology research reproducible enough?" to *eLife*. Your article has been seen by two reviewers and their reports are below.

I asked the reviewers to comment on the following:

# Does the manuscript have any obvious weaknesses?

# Are there any topics you think the manuscript should mention but does not?

# Are there any topics that are covered at too great a length and could be shortened?

# Are there any parts of the article that need to be clearer?

Can you please revise your manuscript to address the comments in the reports and return the revised article to me, with a point-by-point response.

*Reviewer 1:*

Thanks for the opportunity to review the manuscript. We have both enjoyed reading it and think it makes important points. Though we generally agree with the conclusions provided by the authors, we have made a few remarks and suggestions below.

# Does the manuscript have any obvious weaknesses?

1. Our major concern with the manuscript is that the authors’ whole argument, although quite interesting from a theoretical point of view, is based on viewing preclinical research as a diagnostic machine. This might be a good analogy in some cases, when preclinical work corresponds more directly to the subsequent clinical research in its experimental design – e.g. one tests an intervention on animals, and depending on the result, you move to clinical trials with humans.

However, this does not seem to be the case for most preclinical research. Judging by their abstracts, the research within the RP:CB papers seems to be quite variable in terms of its proximity to clinical translation. Some articles are indeed about specific targets, or even about the efficacy of cancer drugs. However, others are about features of specific tumors or understanding signaling pathways. Thus, they are not all "preclinical" in the more strict sense of testing interventions in experimental models.

The authors do recognize this fact in isolated passages, but their reasoning seems to generally ignore the caveat. In their defense, they argue that “probably every sponsor funding these efforts was primarily motivated by the prospects of advancing health”. This is probably true, but the mapping from preclinical finding to clinical advancement is far from linear. One preclinical hypothesis could serve as a basis for many different interventions, especially if it's more fundamental research, further away from the intervention. Conversely, many preclinical findings might have to sum up to produce a candidate for clinical intervention.

This renders the discussion of positive and negative predictive values – which would require one-to-one mapping between preclinical finding and a clinical intervention – and the cost-benefit discussion that follows from it a bit artificial and overly simplistic. There might be no way around this, but the authors could be more clear on the fact that their model represents a crude approximation of a much more nuanced reality. For example, sentences such as “the RPCB results suggest that outputs for this “diagnostic machine” recommend advancing many non-reproducible findings to clinical testing” could probably be avoided in the absence of evidence that these recommendations are being made explicitly in the replicated articles.

[Note from editor: I think the best way to address this point is, as the referee suggests, to acknowledge that the diagnostic machine approach will work for some but not all kinds of preclinical research]

2. An associated problem is that the authors seem to equate citation impact and acceptance in prestigious journals (the basis for the selection of the papers included in RP:CB) with results warranting clinical investigation (e.g. “as specified by their sampling approach (and similar to other replication studies), the RPCB only selected “very positive” outputs from our diagnostic machine.”). Nevertheless, we are not sure that editors and reviewers indeed operate with a “diagnostic machine” mindset, and they could be very well selecting the papers on the basis of other issues (e.g. thoroughness, novelty, theoretical insight, prestige of the research group). Although we certainly agree that high-impact journals will tend to select for more positive findings, calling them all “very positive” in terms of clinical potential on the basis of publication venue and citation impact seems quite a leap.

In the same vein, we do not agree with passages such as “one imperfect way of going about assessing the NPV would be to replicate a sample of low impact original studies” or “another would be to trace the frequency and kinetics with which original reports bounced from high impact journals produce clinical hypotheses that are later vindicated in clinical trials”. Equating rejection at a high-impact journal (or publication in a low-impact venue) with findings being less promising in terms of direct clinical impact (i.e. a “negative” study from the point of view of the diagnostic machine) seems like a very crude and likely inaccurate approximation: there are certainly many positive findings which do not make their way to high-impact journals for other reasons, including low methodological rigor (which is likely to yield false-positive findings on its own).

3. The “intention-to-treat” analysis proposed by the authors seems to suggest that articles for which replication could not be completed should be taken as likely not to reproduce – on the basis of postulations such as “laboratories that were more confident of the reproducibility of their original publications, or more fastidious with record keeping, would be more likely to cooperate with RPCB investigators than laboratories that were more doubtful or less fastidious.” Although this is a possibility, as far as I understood cooperation from the original investigators was not a sine qua non for replications to be carried forward in the RP:CB. Similarly, not all unfinished replications fell by the wayside because of methodological troubles or lack of cooperation: many of them were left out merely because of budget concerns. The authors should thus discuss whether considering attrition as informative really makes sense – if correlation between attrition and non-reproducibility is low, the low predictive value of the intention-to-treat analysis might be largely exaggerating the problem.

# Are there any topics you think the manuscript should mention but does not?

1. As the authors seem to be quite concerned about false negatives and negative predictive values, one discussion that seems to be missing is about publication bias. When negative findings simply go missing (which is probably the case for many of them), estimating negative predictive values is basically impossible, so it is hard to discuss the potential accuracy of the diagnostic machine without taking this issue (i.e. where do negative findings actually end up and what fraction of them is accessible to the “machine”) into consideration. If the authors are concerned about negative predictive values (as they rightly should), ensuring that negative findings are indeed accessible seems like the most pressing issue to solve (and discuss).

2. In the “decision rules” session, the authors seem to suggest that there is a direct tradeoff between sensitivity and specificity. This is indeed the case when one uses a modifiable, quantitative threshold (such as a p value of 0.05) to analyze unbiased data. But is it the case that all false-positives arise because of “loose detection thresholds” considering weak evidence as positive – instead of the threshold itself actually biasing analyses to turn out positive (e.g. by p-hacking and other practices)? For instance, suppose that one or more RP:CB false-positives arose from fraud. If data were forged, changing such a threshold would make little difference (i.e. results would still be positive no matter where the threshold stands).

3. Also in the “decision rules” section, the authors should probably discuss more explicitly that error control for the “diagnostic machine” purposes does not necessarily need to be implemented at the publication level. Type I error rates could also be adjusted at the level of data synthesis – i.e. if companies require more than a few studies in order to move forward with human trials, one could eventually have acceptable error control even if positive predictive value is low at the level of the individual study (which is ultimately what the authors are discussing when referring to RP:CB). This possibility (which is hinted at in the conclusions) should at least be touched upon as an alternative for the diagnostic machine’s decision rules.

4. In some parts of the paper – particularly in the “beliefs” section, the authors engage in a lot of speculation on how the “diagnostic machine process” actually works in real life – e.g. what kind of heuristics companies actually use as evidence to launch a clinical trial. Although speculation is inevitable, as this process has not been charted systematically, it would be interesting if they could provide a clear list (perhaps in a Box 2) of data that could be collected to inform their model and drive the discussion forward.

5. Similarly, when authors make recommendations such as “Preclinical studies should generally be labeled as exploratory, and before advancing a hypothesis into clinical development (…) regulators, sponsors and ethics committees should expect that key preclinical studies be replicated in confirmatory studies where hypotheses and protocols are pre-specified and pre-registered and proper statistical and experimental methods (e.g. randomization) are used”, it might be worth discussing to what extent this might already be happening (or not) in some instances, or even provide examples of existing practices in specific agencies or committees, in case the authors are aware of any.

# Are there any topics that are covered at too great a length and could be shortened?

1. For the general reader, we’re not sure if likelihood ratios (which are an unfamiliar concept to most researchers in the life sciences) are necessary to understand the article’s points. Negative and positive predictive values are more intuitive and I get the feeling that these can largely be used to make the authors’ point without introducing an additional mathematical concept.

# Are there any parts of the article that need to be clearer?

1. The description of the RP:CB methodology is scattered through the paper (i.e. description of study selection and registered reports on page 2, description of the replication methodology on pages 4-5). Wouldn’t it be clearer to describe the methodology in a single session?

[Note from editor: I can deal with this point during editing]

2. The authors mention replication rates in different areas (“To put this number in context, replication rates for psychology were (depending on how you count) 40%, for economics they were 66%, and for the social sciences they were 67%.”) without mentioning the study selection process in each of these studies (i.e. the psychology and economy numbers come from specific specialty journals, the social sciences one from Science and Nature papers). It should be made clear that, as they refer to very specific samples of articles, these replication rates cannot be taken as representative of the whole scientific field.

3. The authors’ use of “inferential reproducibility” in the manuscript does not seem to be in line with the concept’s definition by Goodman et al. (cited to support it as ref. # 23). In that article, “inferential reproducibility” refers to whether two researchers performing similar studies or analyzing the same data will reach similar conclusions on the claims that can be made. This is quite different from “the ability of experts to predict if, how and to what experimental results will generalize” as defined in the current manuscript. Both are important topics to touch on, but they are very different concepts, and a different term should be used to describe the ability to predict the reproducibility of results.

4. When the authors perform calculations to illustrate their point, they assume a few numbers for prevalence rate and negative predictive values (i.e. “For the sake of simplicity, let’s assume that, across the field of cancer research, the “prevalence” or “prior” of credible cancer research hypotheses being sufficiently true (that is, close enough to being true that any adjustments to the hypothesis can be made during clinical development) is 10%. Let’s put an upper bound on the NPV of 99%, that is 99% of negatives are true negatives.”). As we don’t really have a clue on what these numbers really are, it is possible that assuming exact numbers – even for the sake of argument as the authors do – might give the idea of a nonexistent precision to the reader. If the authors do want to make the point – which we agree with – that different parameters lead to very different predictive values and likelihood ratios, it's worth considering whether a figure showing how this varies continuously depending on a wide range of parameter assumptions, without focusing on specific numbers, could be a better way to convey it.

*Reviewer 2:*

Kimmelman and Kane provide an excellent comment on the RPCB. They address three important issues (negative predictive value, decisions, and beliefs) in preclinical research that shape the environment where research is conducted. They also conduct a simple but very informative meta analysis on effect size shrinkage in the RPCB. I personally would be keen on knowing how much shrinkage there was on average as this would be a first prior for future sample size calculations in cancer biology. Perhaps this is also too much to ask at this point or is covered in other meta analyses by Tim Errington and colleagues. I give some more specific comments below that the authors may want to consider.

1. There are some references that already describe a diagnostic machine and make concrete examples. These could be added as they make several suggestions for criteria that could be diagnostic for decisions in this machinery.

- Emmerich et al. 2021 Improving target assessment in biomedical research: the GOT-IT recommendations. Nature Reviews Drug Discovery 20:64–81

– Drude et al. 2021 Improving preclinical studies through replications. *eLife* 10:e62101

(Note: I am a co-author on Drude et al., so please only cite it if you find it helpful.)

[Note from editor: Whether or not you cite either of this references, please mention the reproducibility projects in Brazil and Germany that are mentioned in Drude et al]

2. The authors state: “Experiments that require more finicky conditions seem more likely to require multiple attempts at the bench- thus increasing the prospect of original authors selecting those individual experimental results most flattering to the underlying hypothesis, a problem akin to phenomena elsewhere characterized in reproducibility studies as "researcher degrees of freedom."”

I find this not nuanced enough. Some techniques are only for specialised laboratories and advance our understanding of patho-mechanisms a lot. Examples are two-photon microscopy, organoids and their specific growth factors, single cell sequencing, etc. Not all information can be in the Materials and methods part. Alternative routes of detailed method dissemination are necessary and need higher awareness (*eLife* already promotes these). As it is it may read as if all complicated preclinical research is selective and entirely under researcher degrees of freedom.

3. High Impact Preclinical Experiments are Very Biased Against the Null

It may be helpful here to refer to Colquhoun, D. An investigation of the false discovery rate and the misinterpretation of p-values. Royal Society Open Science 1, 140216. Figure 7 here captures exactly the effect observed in the RPCB.

4. Against Despair, Part II: Decision Rules

The question for me is, was the RPCB designed to enable decision towards translation or was it designed to decide whether evidence is robust in the field of cancer biology. These are two different questions. It should be stated clearly which of these decisions RPCB was designed to make (the authors give a vague answer at the beginning of their comment).

For the question of translation, a field specific approach is necessary that takes all the variables into account that the authors refer to (prior probability, effect sizes in the field, ppv and npv, etc.). I think this question can be found through consensus in a field by combining expert knowledge and meta research and is not so much dependent on morality and sociological conditions as the authors claim in their last paragraph. These two words carry so much and at the same time little specific meaning here that I suggest to tone it down at this point.

5. Some mathematical descriptions could be a bit daunting for the average reader (e.g. paragraph after Box1). To help with that, calculations at the end of the paper could use a figure. It may be easier to understand if they do something similar to Figure 1 in Forstmeier W, Wagenmakers EJ, Parker TH. Detecting and avoiding likely false-positive findings – a practical guide. Biological Reviews of the Cambridge Philosophical Society. 2017 Nov;92(4):1941-1968. DOI: 10.1111/brv.12315.

---

## [Author Response]

Reviewer 1:Thanks for the opportunity to review the manuscript. We have both enjoyed reading it and think it makes important points. Though we generally agree with the conclusions provided by the authors, we have made a few remarks and suggestions below.# Does the manuscript have any obvious weaknesses?1. Our major concern with the manuscript is that the authors’ whole argument, although quite interesting from a theoretical point of view, is based on viewing preclinical research as a diagnostic machine. This might be a good analogy in some cases, when preclinical work corresponds more directly to the subsequent clinical research in its experimental design – e.g. one tests an intervention on animals, and depending on the result, you move to clinical trials with humans.However, this does not seem to be the case for most preclinical research. Judging by their abstracts, the research within the RP:CB papers seems to be quite variable in terms of its proximity to clinical translation. Some articles are indeed about specific targets, or even about the efficacy of cancer drugs. However, others are about features of specific tumors or understanding signaling pathways. Thus, they are not all "preclinical" in the more strict sense of testing interventions in experimental models.The authors do recognize this fact in isolated passages, but their reasoning seems to generally ignore the caveat. In their defense, they argue that “probably every sponsor funding these efforts was primarily motivated by the prospects of advancing health”. This is probably true, but the mapping from preclinical finding to clinical advancement is far from linear. One preclinical hypothesis could serve as a basis for many different interventions, especially if it's more fundamental research, further away from the intervention. Conversely, many preclinical findings might have to sum up to produce a candidate for clinical intervention.This renders the discussion of positive and negative predictive values – which would require one-to-one mapping between preclinical finding and a clinical intervention – and the cost-benefit discussion that follows from it a bit artificial and overly simplistic. There might be no way around this, but the authors could be more clear on the fact that their model represents a crude approximation of a much more nuanced reality. For example, sentences such as “the RPCB results suggest that outputs for this “diagnostic machine” recommend advancing many non-reproducible findings to clinical testing” could probably be avoided in the absence of evidence that these recommendations are being made explicitly in the replicated articles.[Note from editor: I think the best way to address this point is, as the referee suggests, to acknowledge that the diagnostic machine approach will work for some but not all kinds of preclinical research]

We agree that our model and theory (like any model or theory) strips away complexity and nuance to get traction on a problem; some studies in the RPCB are indeed more basic than others. We also appreciate that the assumptions and implications embedded in our model were not as explicit as they could have been and could lead to confusion. Accordingly, we added a “provisos and clarifications” paragraph to the section that introduces the model that addresses three aspects of the model flagged by this referee- (a) the imperfection of the model; (b) the way diagnostic machine outputs might be used by different actors; (c) the precise meaning of what “positive” and “negative” means:

“Some clarifications and provisos before proceeding with our analysis. First, we acknowledge that analogizing cancer biology research as a diagnostic machine does not accommodate the spirit and goals of all original reports included in the RPCB. Like all models, ours strips away complexity to bring essential elements of a system to the fore. Nevertheless, every sponsor funding these efforts was probably motivated by the prospects of advancing health. Even the most basic findings of studies in the RPCB have some prospect of being advanced in some fashion to clinical applications. Second, we recognize that positive outputs of the diagnostic machine will be used differently. In some cases, they might be grounds to launch a clinical trial. In other cases, they will promote a clinical hypothesis to further work on a critical path towards translation – perhaps by supporting development of more pharmacologically attractive compounds that target the causal process fished out in an original report. Third, in our analogy, outputs of the diagnostic machine are to be understood not as epistemic outputs (e.g. positive outputs truly signal a hypothesis is promising; negative the reverse). Instead the outputs are to be understood as sociological- positive outputs signal that expert communities regard a hypothesis as worthy of further advancement. In this way, our analogy regards the RPCB as determining whether repetition of key experiments in original reports would have produced results consistent with previous ones that generated a buzz in the cancer biology community. In a well-functioning research system, sociological “truths” will track epistemic truths. However, the former will also be informed by nonepistemic variables, including views about clinical need. In the next sections, we consider what the RPCB results tell us about the performance of this sprawling clinical hypothesis diagnostic machine (Box 1 provides a primer on terminology that will be used in our analysis).”

We also clarify that we understand the process of medical translation is not about developing molecules per se, but rather unlocking the clinical utility of concepts and strategies that basic and preclinical research uncovers (that the first anti-angiogenesis compound, endostatin, failed clinical development does not take away from the enormous success of the paradigm in contemporary cancer management). This is reflected in the following passage (italics indicating text insertion):

“If individual experiments within each original publication converge on the same translational claim (which may not be a purely material claim about a particular molecule, but a claim about strategy) and produce large effects, the diagnostic machine recommends clinical development. If individual experiments point in inconsistent directions, if individual experiments effects are small, or if quality control flags a study, the diagnostic machine recommends against clinical development- at least for the time being.”

2. An associated problem is that the authors seem to equate citation impact and acceptance in prestigious journals (the basis for the selection of the papers included in RP:CB) with results warranting clinical investigation (e.g. “as specified by their sampling approach (and similar to other replication studies), the RPCB only selected “very positive” outputs from our diagnostic machine.”). Nevertheless, we are not sure that editors and reviewers indeed operate with a “diagnostic machine” mindset, and they could be very well selecting the papers on the basis of other issues (e.g. thoroughness, novelty, theoretical insight, prestige of the research group). Although we certainly agree that high-impact journals will tend to select for more positive findings, calling them all “very positive” in terms of clinical potential on the basis of publication venue and citation impact seems quite a leap.

Thank you for pressing us to be more precise. I don’t think we disagree. We intend “positive” in the broadest sense possible, namely that claims embedded in publications are deemed by the expert community to have some cluster of qualities (e.g. excitement, reliability, impact) that warrants the visibility bestowed by prestigious journals. We have clarified this in the paragraph addressing the referee’s first concern.

In the same vein, we do not agree with passages such as “one imperfect way of going about assessing the NPV would be to replicate a sample of low impact original studies” or “another would be to trace the frequency and kinetics with which original reports bounced from high impact journals produce clinical hypotheses that are later vindicated in clinical trials”. Equating rejection at a high-impact journal (or publication in a low-impact venue) with findings being less promising in terms of direct clinical impact (i.e. a “negative” study from the point of view of the diagnostic machine) seems like a very crude and likely inaccurate approximation: there are certainly many positive findings which do not make their way to high-impact journals for other reasons, including low methodological rigor (which is likely to yield false-positive findings on its own).

As we now discuss in our section on provisos and clarification (see above), there are two broad ways one might understand outputs of the diagnostic machine in our analogy. One is in an epistemic sense: positive outputs tell us what is truly a clinically impactful discovery. The other is in a sociological sense: positive outputs tell us what the machinery of science (which combines scientific processes, like the performance of assays, with sociological ones like peer review) tell us is an actionable discovery. Because we generally think science is well organized, we assume that latter will track the former, if imperfectly. In our analogy, we understand the diagnostic machine to be giving us a read out of the latter. What the RPCB is doing is activating those same machinery of scientific research and seeing if it produces results that are consistent with original findings that excited research communities (i.e., individual experimental results with large effect sizes, which when assembled together, offer a coherent picture about a causal process in cancer). To be clear, it is entirely possible, indeed likely, that many “positive” and “negative” findings are wrong in the epistemic sense. Some “positive” outputs might be artifacts of animal models. Some “negative” outputs might similarly reflect use of animal models that do not recapitulate human pathophysiology. However, we regard this mismatch between models and reality as a problem that is intrinsic to science itself.

As a result, when we say that studies published in low impact venues are less promising in terms of advancing cancer treatment, we do not mean that their findings are truly less important for developing cancer treatments. Instead, we mean that they are less likely to be interpreted by expert communities as likely to impact cancer care.

The idea behind our proposal is asking whether experiments that do not excite expert communities (hence published in lower impact venues), when repeated, acquire properties (i.e. individual experimental results with large effect sizes, which when assembled together, offer a coherent picture about a causal process in cancer) that excite expert communities.

3. The “intention-to-treat” analysis proposed by the authors seems to suggest that articles for which replication could not be completed should be taken as likely not to reproduce – on the basis of postulations such as “laboratories that were more confident of the reproducibility of their original publications, or more fastidious with record keeping, would be more likely to cooperate with RPCB investigators than laboratories that were more doubtful or less fastidious.” Although this is a possibility, as far as I understood cooperation from the original investigators was not a sine qua non for replications to be carried forward in the RP:CB. Similarly, not all unfinished replications fell by the wayside because of methodological troubles or lack of cooperation: many of them were left out merely because of budget concerns. The authors should thus discuss whether considering attrition as informative really makes sense – if correlation between attrition and non-reproducibility is low, the low predictive value of the intention-to-treat analysis might be largely exaggerating the problem.

We agree that it is impossible to know whether incomplete studies are less reproducible than ones that completed. We do, however, feel confident in saying that studies the RPCB were unable to repeat cannot be assumed to be missing at random, and we think there are numerous plausible reasons to entertain the possibility that completed studies are biased towards reproducibility. We have revised the manuscript to clarify that this intention to treat analysis is an attempt to create a lower bound, rather than the single correct estimate:

“The remaining 42 replication studies did not, thus providing a lower bound on the positive predictive value of 16%- an estimate that is not far off from the proportion of studies Amgen reported reproducing years ago.(1)”

# Are there any topics you think the manuscript should mention but does not?1. As the authors seem to be quite concerned about false negatives and negative predictive values, one discussion that seems to be missing is about publication bias. When negative findings simply go missing (which is probably the case for many of them), estimating negative predictive values is basically impossible, so it is hard to discuss the potential accuracy of the diagnostic machine without taking this issue (i.e. where do negative findings actually end up and what fraction of them is accessible to the “machine”) into consideration. If the authors are concerned about negative predictive values (as they rightly should), ensuring that negative findings are indeed accessible seems like the most pressing issue to solve (and discuss).

We understand publication bias in the above to refer to circumstances where individual experiments are strung together into a potential original report, but because the potential original report is null, it is not advanced toward publication. So instead of showing up as a “low impact” publication, it stays in the file drawer. We agree this type of publication bias poses a major practical challenge for assessing NPV. We now acknowledge publication bias as presenting a challenge for two approaches we outline for assessing NPV by a) stating that “many negative findings are… never published (according to one estimate, only 58% of animal studies are eventually published).”(2) and (b) by adding the sentence “Both of these options would be limited by sampling bias, since many negative studies are never submitted for publication.”

2. In the “decision rules” session, the authors seem to suggest that there is a direct tradeoff between sensitivity and specificity. This is indeed the case when one uses a modifiable, quantitative threshold (such as a p value of 0.05) to analyze unbiased data. But is it the case that all false-positives arise because of “loose detection thresholds” considering weak evidence as positive – instead of the threshold itself actually biasing analyses to turn out positive (e.g. by p-hacking and other practices)? For instance, suppose that one or more RP:CB false-positives arose from fraud. If data were forged, changing such a threshold would make little difference (i.e. results would still be positive no matter where the threshold stands).

We respectfully disagree with the assertion that sensitivity and specificity don’t trade-off more generally. While there may be some areas in which a free lunch can be found, at least in the area of fraud, any enforcement mechanism used to detect fraudulent results will have its own false positives and false negatives, essentially recapitulating the same dynamic that applies to Gaussian curves.

3. Also in the “decision rules” section, the authors should probably discuss more explicitly that error control for the “diagnostic machine” purposes does not necessarily need to be implemented at the publication level. Type I error rates could also be adjusted at the level of data synthesis – i.e. if companies require more than a few studies in order to move forward with human trials, one could eventually have acceptable error control even if positive predictive value is low at the level of the individual study (which is ultimately what the authors are discussing when referring to RP:CB). This possibility (which is hinted at in the conclusions) should at least be touched upon as an alternative for the diagnostic machine’s decision rules.

We partly agree. In the previous version, we note that regulatory systems intercept false positives and thus reduce the “loss” associated with them. We revised this sentence to include other quality control processes (italics indicate additions):

“So long as we have mechanisms for intercepting false positives, such as effective drug regulators or quality control processes in research, errant acceptance of a clinical hypothesis will be quickly spotted in clinical testing.”

Our piece previously noted the importance of embedding new findings against what is known about a hypothesis; in revisions, we now also describe pathways that sponsors might consider in verifying strategies before putting them into clinical trials (see below).

On the other hand, we also think our point still stands. If *Nature* conditioned paper acceptance of all submitted papers on authors performing replications, it would have the same effect of upping the p-value. Namely, more resources would be expended per ‘buzz generating publication,’ with the consequence that some truly promising findings are missed because effect sizes in repeat experiments were insufficiently large (due to chance) for *Nature* to accept the piece. In many ways, the spirit of this comment is addressed in the next section (under “Beliefs”).

4. In some parts of the paper – particularly in the “beliefs” section, the authors engage in a lot of speculation on how the “diagnostic machine process” actually works in real life – e.g. what kind of heuristics companies actually use as evidence to launch a clinical trial. Although speculation is inevitable, as this process has not been charted systematically, it would be interesting if they could provide a clear list (perhaps in a Box 2) of data that could be collected to inform their model and drive the discussion forward.

We weren’t sure how to integrate these suggestions into our text without disturbing the flow. We added a few sentences to the end of the beliefs section that lays out a research agenda for better understanding the belief dimensions of drug development.

5. Similarly, when authors make recommendations such as “Preclinical studies should generally be labeled as exploratory, and before advancing a hypothesis into clinical development (…) regulators, sponsors and ethics committees should expect that key preclinical studies be replicated in confirmatory studies where hypotheses and protocols are pre-specified and pre-registered and proper statistical and experimental methods (e.g. randomization) are used”, it might be worth discussing to what extent this might already be happening (or not) in some instances, or even provide examples of existing practices in specific agencies or committees, in case the authors are aware of any.

This request echoes suggestions made by referee 2. We now cite two sources that describe pathways for validation of strategies before clinical development (Emmerich et al; Drude et al).

# Are there any topics that are covered at too great a length and could be shortened?1. For the general reader, we’re not sure if likelihood ratios (which are an unfamiliar concept to most researchers in the life sciences) are necessary to understand the article’s points. Negative and positive predictive values are more intuitive and I get the feeling that these can largely be used to make the authors’ point without introducing an additional mathematical concept.

We appreciate this concern. However, we believe these concepts are important (and widespread) enough to be worth incorporated in the manuscript. We’ve done our best to provide context and explanation- and for the motivated reader- citations where they can follow up. Perhaps the new figure will at least help readers visually connect the relationship between PPV, priors, and strength of evidence.

# Are there any parts of the article that need to be clearer?1. The description of the RP:CB methodology is scattered through the paper (i.e. description of study selection and registered reports on page 2, description of the replication methodology on pages 4-5). Wouldn’t it be clearer to describe the methodology in a single session?[Note from editor: I can deal with this point during editing]

Great- thanks.

2. The authors mention replication rates in different areas (“To put this number in context, replication rates for psychology were (depending on how you count) 40%, for economics they were 66%, and for the social sciences they were 67%.”) without mentioning the study selection process in each of these studies (i.e. the psychology and economy numbers come from specific specialty journals, the social sciences one from Science and Nature papers). It should be made clear that, as they refer to very specific samples of articles, these replication rates cannot be taken as representative of the whole scientific field.

Thank you for noting this we have now provided more context on the journals involved in these replication projects.

3. The authors’ use of “inferential reproducibility” in the manuscript does not seem to be in line with the concept’s definition by Goodman et al. (cited to support it as ref. # 23). In that article, “inferential reproducibility” refers to whether two researchers performing similar studies or analyzing the same data will reach similar conclusions on the claims that can be made. This is quite different from “the ability of experts to predict if, how and to what experimental results will generalize” as defined in the current manuscript. Both are important topics to touch on, but they are very different concepts, and a different term should be used to describe the ability to predict the reproducibility of results.

Thank you for catching this; we regret the error. We removed the reference.

4. When the authors perform calculations to illustrate their point, they assume a few numbers for prevalence rate and negative predictive values (i.e. “For the sake of simplicity, let’s assume that, across the field of cancer research, the “prevalence” or “prior” of credible cancer research hypotheses being sufficiently true (that is, close enough to being true that any adjustments to the hypothesis can be made during clinical development) is 10%. Let’s put an upper bound on the NPV of 99%, that is 99% of negatives are true negatives.”). As we don’t really have a clue on what these numbers really are, it is possible that assuming exact numbers – even for the sake of argument as the authors do – might give the idea of a nonexistent precision to the reader. If the authors do want to make the point – which we agree with – that different parameters lead to very different predictive values and likelihood ratios, it's worth considering whether a figure showing how this varies continuously depending on a wide range of parameter assumptions, without focusing on specific numbers, could be a better way to convey it.

Thanks for this helpful suggestion. We have prepared a figure to address this comment.

Reviewer 2:Kimmelman and Kane provide an excellent comment on the RPCB. They address three important issues (negative predictive value, decisions, and beliefs) in preclinical research that shape the environment where research is conducted. They also conduct a simple but very informative meta analysis on effect size shrinkage in the RPCB. I personally would be keen on knowing how much shrinkage there was on average as this would be a first prior for future sample size calculations in cancer biology. Perhaps this is also too much to ask at this point or is covered in other meta analyses by Tim Errington and colleagues. I give some more specific comments below that the authors may want to consider.1. There are some references that already describe a diagnostic machine and make concrete examples. These could be added as they make several suggestions for criteria that could be diagnostic for decisions in this machinery.- Emmerich et al. 2021 Improving target assessment in biomedical research: the GOT-IT recommendations. Nature Reviews Drug Discovery 20:64–81– Drude et al. 2021 Improving preclinical studies through replications. eLife 10:e62101(Note: I am a co-author on Drude et al., so please only cite it if you find it helpful.)[Note from editor: Whether or not you cite either of this references, please mention the reproducibility projects in Brazil and Germany that are mentioned in Drude et al]

Thank you for these suggestions; we had not been aware of either of these publications and appreciate being alerted to each. We now cite the Emmerich et al. and Drude et al. in our closing section: “Emmerich et al. offer a pathway for assessing new strategies for clinical development that is informed by validity threats in preclinical research [EMMERICH et al] As other commentators have urged, it encourages researchers to label preclinical studies as exploratory.[DRUDE et al]”

2. The authors state: “Experiments that require more finicky conditions seem more likely to require multiple attempts at the bench- thus increasing the prospect of original authors selecting those individual experimental results most flattering to the underlying hypothesis, a problem akin to phenomena elsewhere characterized in reproducibility studies as "researcher degrees of freedom."”I find this not nuanced enough. Some techniques are only for specialised laboratories and advance our understanding of patho-mechanisms a lot. Examples are two-photon microscopy, organoids and their specific growth factors, single cell sequencing, etc. Not all information can be in the Materials and methods part. Alternative routes of detailed method dissemination are necessary and need higher awareness (eLife already promotes these). As it is it may read as if all complicated preclinical research is selective and entirely under researcher degrees of freedom.

We agree that some techniques might require specialized labs. However, Errington et al. describe how they attempted to minimize failed replications due to the use of specialized techniques: “[we excluded studies] if they required specialized samples, techniques, or equipment that would be difficult or impossible to obtain,” and “a key part of matching experiments with laboratories is to identify labs with the appropriate expertise to maximize research quality. This is particularly important with new and innovative techniques, though most techniques called for in the selected experiments are standard techniques for which expertise is widely avail- able. As experiments are matched to labs, it is possible that no appropriate service provider can be identified. If appropriate expertise is not available, then the finding or paper will be excluded from the project.” We nevertheless acknowledge that RPCB might have erred in its judgments- now stated in revisions (italics indicating new text):

“RPCB states that they selected experiments that didn’t rely on unusual samples or techniques, and they attempted to match experiments to laboratories possessing competency with methods [ERRINGTON 2014]. Acknowledging the possibility RPCB selection and matching might have erred, studies involving standardized experimental techniques would be likely easier to reproduce than studies using less familiar ones, since there are likely to be more quality controls on standardized reagents.”

3. High Impact Preclinical Experiments are Very Biased Against the NullIt may be helpful here to refer to Colquhoun, D. An investigation of the false discovery rate and the misinterpretation of p-values. Royal Society Open Science 1, 140216. Figure 7 here captures exactly the effect observed in the RPCB.

Many thanks. Now cited in this section- also in the closing section.

4. Against Despair, Part II: Decision RulesThe question for me is, was the RPCB designed to enable decision towards translation or was it designed to decide whether evidence is robust in the field of cancer biology. These are two different questions. It should be stated clearly which of these decisions RPCB was designed to make (the authors give a vague answer at the beginning of their comment).For the question of translation, a field specific approach is necessary that takes all the variables into account that the authors refer to (prior probability, effect sizes in the field, ppv and npv, etc.). I think this question can be found through consensus in a field by combining expert knowledge and meta research and is not so much dependent on morality and sociological conditions as the authors claim in their last paragraph. These two words carry so much and at the same time little specific meaning here that I suggest to tone it down at this point.

Thank you for these observations and suggestions. We agree consensus in a field can be found by combining expert knowledge and meta-research. But note that combining expert knowledge (and meta-research, for that matter) is itself a sociological exercise. To the former, we must begin by demarcating experts from nonexperts- which right off the bat is a sociological exercise. And we assume that part of what gives an individual their expertise involves both their commitment to certain values as well as their knowledge of the value environment in which they are working. For example, physician experts should have particular values (e.g. they should value a duty to not harm their patients, say) and they should also have an accurate read on what their patients value (what side effects they can live with; what disabilities they can’t abide). So, we stand by our core point, though we regret that elaborating on this might sidetrack the reader unduly and have thus elected to leave this point unchanged.

To the point at the start of this comment (objectives of the RPCB) we hope our revisions on the front end of the piece (section 2) go some way to addressing this.

5. Some mathematical descriptions could be a bit daunting for the average reader (e.g. paragraph after Box1). To help with that, calculations at the end of the paper could use a figure. It may be easier to understand if they do something similar to Figure 1 in Forstmeier W, Wagenmakers EJ, Parker TH. Detecting and avoiding likely false-positive findings – a practical guide. Biological Reviews of the Cambridge Philosophical Society. 2017 Nov;92(4):1941-1968. DOI: 10.1111/brv.12315.

Thank you for this suggestion and the citation. We’ve seen similar figures elsewhere- I believe in the Economist (or, fivethirtyeight.com). If the editors advise, we’d be happy to prepare such a figure; otherwise we now cite a paper in Nature (Mogul / Macleod) that uses this same graphic approach, and hope the figure we prepared in response to referee adds an additional graphical representation of our conclusions.

References

1) Begley CG, Ellis LM. Drug development: Raise standards for preclinical cancer research. Nature. 2012 Mar;483(7391):531–3.

2) Wieschowski S, Biernot S, Deutsch S, Glage S, Bleich A, Tolba R, et al. Publication rates in animal research. Extent and characteristics of published and non-published animal studies followed up at two German university medical centres. PLoS One [Internet]. 2019 Nov 26 [cited 2021 Jun 4];14(11). Available from: https://www.ncbi.nlm.nih.gov/pmc/articles/PMC6879110/